



**An assessment of land energy balance over East Asia from**
**multiple lines and the roles of Tibet Plateau, aerosols, and**
**clouds**
Qiuyan Wang[1,2,6], Hua Zhang[1,2], Su Yang[3], Qi Chen[2], Xixun Zhou[2], Bing Xie[4], Yuying
Wang[1], Guangyu Shi[1,5], Martin Wild[6]
[1] Collaborative Innovation Center on Forecast and Evaluation of Meteorological Disasters, Nanjing University
of Information Science and Technology, Nanjing 210044, China
[2] State Key Laboratory of Severe Weather, Chinese Academy of Meteorological Sciences, Beijing 100081,
China
[3] National Meteorological Information Center, China Meteorological Administration, Beijing 100081, China
[4] Laboratory for Climate Studies of China Meteorological Administration, National Climate Center, Beijing
100081, China
[5] State Key Laboratory of Numerical Modeling for Atmospheric Sciences and Geophysical Fluid Dynamics,
Institute of Atmospheric Physics, Chinese Academy of Sciences, Beijing 100029, China
[6] Institute for Atmospheric and Climate Science, ETH Zurich, 8092 Zurich, Switzerland
*Corresponding to*: Hua Zhang (huazhang@cma.gov.cn)



**Abstract.** With high emissions of aerosols and the known world's "Third Pole" of the Tibet Plateau (TP) in
East Asia, knowledge on the energy budget over this region is widely concerned. This study first attempts
to estimate the present-day land energy balance over East Asia by combining surface and satellite
observations, as well as the atmospheric reanalysis and Coupled Model Intercomparison Project phase 6
(CMIP6) simulations. Compared to the global land budget, a substantially larger fraction of atmospheric
shortwave radiation of 5.2% is reflected, highly associated with the higher aerosol loadings and more clouds
over East Asian land. While a slightly smaller fraction of atmospheric shortwave absorption of 0.6% is
unexpectedly estimated, possibly related to the lower water vapor content effects due to the thinner air over
the TP to overcompensate for the aerosol and cloud effects over East Asian land. The weaker greenhouse
effect and fewer low clouds due to the TP are very likely the causes for the smaller fraction of East Asian-
land surface downward longwave radiation. Hence, high aerosol loadings, clouds, and the TP over East Asia
play vital roles in the shortwave budgets, while the TP is responsible for the longwave budgets during this
regional energy budget assessment. The further obtained cloud radiative effects suggest that the presence of
clouds results in a larger cooling effect on the climate system over East Asian land than that over globe. This
study helps understand the potential factors influencing the diversifying energy budget assessments over
regions.

## 1. Introduction

Current patterns of Earth's weather and climate are largely determined by the spatiotemporal
distributions of energy exchanges between the surface, atmosphere, and space. Theoretically, the outgoing
longwave radiation (OLR) is balanced by the incoming and reflected solar radiation at the top of the
atmosphere (TOA) to produce an equilibrium climate. The incoming solar radiation can be scattered by
clouds and aerosols or absorbed by the intermediary atmosphere, thereby contributing to the diverse energy
transformation at the surface (Trenberth et al., 2009; Wild et al., 2013a). The Earth's surface energy balance
is of particular significance because it is the key driver of atmospheric and oceanic circulations, hydrological
cycles, and various surface processes (Wild et al., 2008; Mercado et al., 2009; Wild et al., 2013a; L Ecuyer
et al., 2015). Anthropogenic influences on climate change are driven by the uneven distribution of the TOA
net radiation caused by forcings perturbed by variations of the atmospheric composition of greenhouse gases
and aerosols as well as aerosol-cloud interactions (Trenberth et al., 2009; Stephens et al., 2012; Wild et al.,
2013a; Trenberth et al., 2014; L Ecuyer et al., 2015; Wild et al., 2019).

Many efforts have been made to quantify the magnitudes of different radiative components or energy
budgets in the climate system over a range of time-space scales, such as on global scales (Lin et al., 2008;
Trenberth et al., 2009; Stephens et al., 2012; Wild et al., 2013b; Wild et al., 2015; L Ecuyer et al., 2015;
Wild et al., 2019; Wild, 2020), over land and ocean domains or the energy transport between them (Fasullo
and Trenberth, 2008a, b; Trenberth et al., 2009; Wild et al., 2015; L Ecuyer et al., 2015), over the Arctic
(Previdi et al., 2015; Christensen et al., 2016), and over individual continents and ocean basins (L Ecuyer et
al., 2015; Kim and Lee, 2018; Thomas et al., 2020). The energy balance at the TOA can be accurately



monitored by satellites from the most advanced Clouds and the Earth's Radiant Energy System (CERES)
Energy Balanced and Filled (EBAF) data product (Loeb et al., 2018), while considerably larger uncertainties
appear at the surface fluxes owing to weaker observational constraints (Raschke et al., 2016; Kato et al.,
2018; Huang et al., 2019). These assessments mostly build upon complementary approaches from a
combination of space and surface observations, climate models, and reanalyses. To date, the discrepancies
of independent global mean surface radiative fluxes have estimated to be within a few W m$^{-2}$ (Wild, 2017a,
b), enabling the accurate quantification of global surface budgets. In addition, the surface radiative
components simulated by various climate models vary substantially in a range of around 10–20 W m$^{-2}$ on
global scales, but exhibit greater inter-model discrepancies on regional scales (Li et al., 2013; Wild et al.,
2013a; Boeke and Taylor, 2016; Wild et al., 2015; Wild, 2017a, b, 2020). Existing challenges on the surface
energy estimates include considerable uncertainties from surface albedo and skin temperature, as well as the
partitioning of surface net radiation into sensible and latent heat (SH; LH) (Wild, 2017a, b).
As the world's largest and highest plateau, the Tibet Plateau (TP) covers nearly one third of the East
Asian land area, significantly affecting the atmospheric circulation, energy budget, and water cycles of
climate system through its orographic and thermal effects (Liu et al., 2007; Xu et al., 2008a, b; Wu et al.,
2015). Deeper insights into the energy budget differences over East Asian and global land under the
background of high aerosol emissions and the role of the TP in East Asia are of the meaningful and essential
attempts. Therefore, our emphasis in this study is on the regional characterization of the East Asian energy
balance under both all-sky and clear-sky conditions based on a combination of surface observations, satellite-
derived products, reanalysis, and Coupled Model Intercomparison Project phase 6 (CMIP6) models. The
cloud influence on the radiative energy budgets at the TOA, within the atmosphere, and at the surface is
further quantified over this region. Section 2 introduces the different data sources used in this study,
including surface and satellite observations, climate models, and reanalysis. Sections 3 and 4 provide detailed
analyses of the all-sky and clear-sky estimates of the energy balance components. The inferred cloud
radiative effects (CREs) at the TOA, within the atmosphere, and at the surface are presented in Section 5.
Summary and conclusions are given in Section 6. The present-day in this study represents years of 2010–
2014, which corresponds to the last five years of the historical simulations in CMIP6 climate models. East
Asian land as considered in this study consists of five countries, including China, Japan, South and North
Korea, as well as Mongolia.

## 2. Data sources

### 2.1. Surface observations

Considering the efforts to diminish the inhomogeneities in the measurement of ground-based surface
(downward) solar radiation (SSR) (Tang et al., 2011; Wang, 2014; Wang et al., 2015; Wang and Wild, 2016;
He et al., 2018; Yang et al., 2018, 2019) and the large amount of observational stations over China, the
homogenized monthly all-sky and clear-sky SSR datasets from the China Meteorological Administration
(CMA) National Meteorological Information Center (NMIC) are used in this study (http://data.cma.cn/enl)



(Yang et al., 2018, 2019). In this dataset, the clear-sky condition at observational sites is defined based on
the measured cloud fraction per day of no more than 15% (Yang et al., 2018). Taking clear-sky data (with
relatively complex missing months compared to the all-sky dataset) as an example, sites with more than one
year of $> 2$ missing months were deleted to ensure $\geq 4$ years of available data during the period 2010-2014,
then the spline interpolation was performed on the missing months of the selected sites. As a consequence,
99 and 76 sites are available for the all-sky and clear-sky studies, respectively. Besides, to further explore
the impacts from different site types, 84 (62) urban and 15 (14) rural stations for all-sky (clear-sky)
conditions are defined according to the administrative divisions of China (Wang et al., 2017).
For the remaining East Asian sites, we use the monthly Global Energy Balance Archive (GEBA) dataset
( http://www.geba.ethz.ch) (Wild et al., 2017), which contains a worldwide widespread distribution of
monthly data from many sources, e.g., from the World Radiation Data Center (WRDC), the Baseline Surface
Radiation Network (BSRN), etc. Among these data sources, the BSRN dataset has a much higher precision
and temporal resolution (up to 1 min) compared to the GEBA, but its site number is very limited over East
Asia (only a few sites located in Japan and one site in Xianghe, China, but with no data available during this
study period), thus making it impossible to obtain clear-sky data using the clear-sky detection algorithm.
Moreover, the relative random error of the monthly SSR from the GEBA data evaluated by Gilgen et al.
(1998) is 5%.
In order to retain as many sites as possible during the study period, we widen the selection criterion of
the GEBA data, i.e., sites with data $\geq 4$ years and missing months $\leq 3$. Eventually, 8, 2, 4, and 14 sites are
selected from GEBA in China, Mongolia, South and North Korea, and Japan, respectively. Especially,
among the 14 sites in Japan, five pairs of the duplicate sites are obtained from the WRDC and BSRN sources,
respectively, and the left 4 sites are only from the WRDC (9 sites available). For China, only one site from
Hongkong out of 8 GEBA sites is not repetitive from the above-mentioned CMA sites (1 site available).
Therefore, 16 out of 28 GEBA sites are available under all-sky conditions (including 15 sites over regions
outside China and 1 site over Hongkong, China) by taking the average of these duplicate sites in Japan
instead, while the clear-sky reference sites are obtained from the interpolated CERES EBAF clear-sky
estimates at the GEBA sites (also 16 sites) due to the limited numbers of observational sites over these
regions. Additionally, we regard four island sites in Japan as rural stations (not shown in the figures), while
the sites in Mongolia as well as South and North Korea are all urban sites.
As shown in Fig. S1, there are 99 (rural/total: 15/99) and 16 (rural/total: 4/16) sites from the CMA and
GEBA available under all-sky conditions, respectively, whereas 76 (rural/total: 14/99) and 16 (from the
CERES-interpolated data at the 16 GEBA sites) sites are considered for clear-sky conditions, respectively.
More detailed station information is given in Table S1.



### 2.2. Satellite observation


Owing to the excellent temporal and spatial coverage of satellite instruments, CERES data products are
widely used to track variations of Earth's energy budgets. The newly released CERES EBAF Edition 4.1
with a monthly 1°×1° latitude-longitude resolution is used in this study (https://ceres.larc.nasa.gov/data/). In
this dataset, the TOA radiation components are adjusted within their uncertainty ranges based on the
independent observational ocean estimates of global heating rate (Loeb et al., 2018). Unlike the directly
measured TOA energy budget, the EBAF-surface energy fluxes are calculated by the cloud and aerosol
properties from satellite-derived products as well as the atmospheric profiles from reanalysis, with a lower
accuracy than their TOA counterparts (Kato et al., 2018). The uncertainty ranges in 1°×1° regional monthly
all-sky and clear-sky longwave (LW) and shortwave (SW) radiation fluxes at the TOA are also documented
by Loeb et al. (2018).

### 2.3. Climate models and reanalysis


Data from 40 CMIP6 climate models are used for the analyses in this study with their model
abbreviations, modeling groups, and resolutions in Table S2. A detailed description of the modeling groups
participating in CMIP6 is provided at https://pcmdi.llnl.gov/CMIP6/. The CMIP6 model-calculated radiation
fluxes under investigation for this study include energy budgets under both all-sky and clear-sky conditions
from 'historical all forcings' experiments covering the period 2010-2014. In these historical simulations,
both natural (e.g., solar variability and volcanic aerosols) and anthropogenic (e.g., greenhouse gases, aerosols,
and land use) forcings are considered to reproduce the climate change and evolution since preindustrial times
as accurately as possible (Eyring et al., 2016). Only the first ensemble member of each model is selected for
the analysis and the model numbers vary slightly among different available energy components.
In the long history of the European Center for Medium-range Weather Forecast (ECMWF), ERA5 is
the fifth generation product. It is a comprehensive reanalysis from 1979 (soon be backdated to 1950) to near
real time, which assimilates as many observations as possible in the upper air and near surface
(https://cds.climate.copernicus.eu/). Monthly means of the radiative components from ERA5 are used in this
study with a resolution of 0.25°×0.25° (regridded to 1°×1°). Compared to previous reanalyses (such as ERA-
Interim), a major strength of ERA5 is the much higher temporal and spatial resolutions, as well as a larger
number of vertical levels (Hersbach et al., 2020). Several independent studies have evaluated the
performance of ERA5 since its release. For example, excellent closure of the Arctic energy budget based on
ERA5 atmospheric data has been assessed by Mayer et al. (2019). The representation of surface irradiance
of ERA5 has been compared with other reanalyses and with ground and satellite observations (Trolliet et al.,
2018; Urraca et al., 2018). Specifically, Trolliet et al. (2018) found that the surface solar irradiance over the
tropical Atlantic Ocean from ERA5 exhibits fewer biases than the second version of the Modern-Era
Retrospective Analysis for Research and Applications (MERRA-2). Urraca et al. (2018) reported that ERA5
can be a valid alternative for satellite-derived products in terms of surface irradiance in most inland stations
compared to ERA-Interim or MERRA-2.



## 3. Assessment of land energy balance budgets under all-sky conditions

### 3.1. Shortwave components

Under all-sky conditions, the present-day annual land-mean anomalies of TOA incident solar radiation as well as the SW net radiation at the TOA, within the atmosphere, and at the surface regarding to their respective multi-model means as simulated by various CMIP6 models over East Asia are shown in Fig. 1a. A summary of the CMIP6 model statistics (such as available model number, model spread, and the standard deviation (SD)), along with the corresponding multi-model mean, ERA5-, and CERES-derived estimates of different energy balance components are listed in Table 1. As shown in Fig. 1a, with the exception of the BCC-CSM2-MR and BCC-CESM1 models, all models give an estimate around 334 W m$^{-2}$ for TOA incoming solar radiation with a very small SD of 0.2, closely matching the multi-model mean as well as the CERES and ERA5 estimates (Table 1). The multi-model means of solar absorption at the TOA, within the atmosphere, and at the surface are 217, 73, and 144 W m$^{-2}$, respectively, all within 2 W m$^{-2}$ of the biases against the CERES-derived estimates, while they are 3–4 W m$^{-2}$ larger for those from ERA5 at the TOA and within the atmosphere, yielding 1 W m$^{-2}$ of bias against the CERES-based estimate at the surface (Table 1). However, the individual models vary significantly in their simulated annual East Asian land-mean solar absorption both at the TOA and surface (Fig. 1a), with SDs of around 6 W m$^{-2}$ and inter-model spreads of more than 20 W m$^{-2}$ (Table 1). Considering the smaller absolute amount of atmospheric and surface solar absorption compared to the TOA counterpart (73 and 144 vs. 217 W m$^{-2}$; Table 1), the relative (percentage) differences relative to their respective multi-model means ( relative (percentage) difference = $\frac{\text{range}}{\text{multi-model mean}} \times 100\%$) indicate that the uncertainties within the atmosphere and at the surface are larger than that at the TOA (i.e., TOA: $\frac{22}{217} \times 100\% = 10\%$; Atmosphere: $\frac{19}{73} \times 100\% = 26\%$; Surface: $\frac{23}{144} \times 100\% = 16\%$).



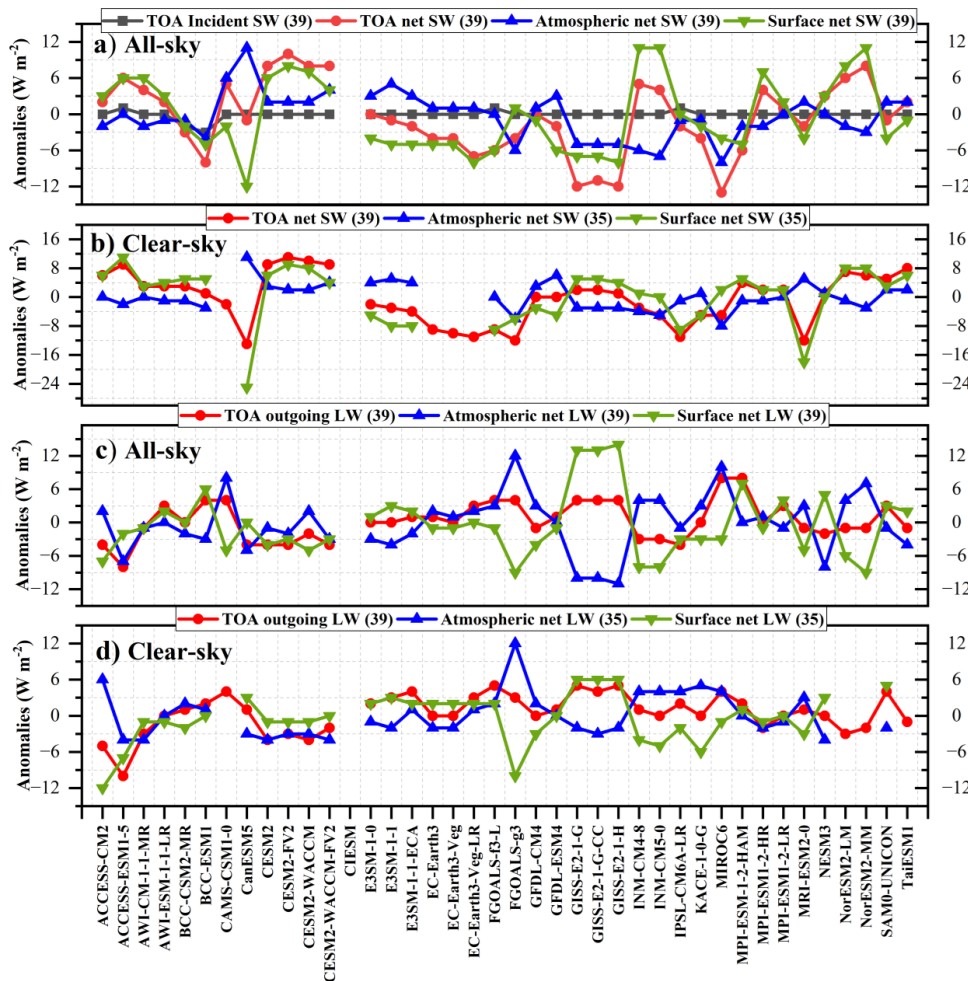

**Figure 1**. Annual land mean anomalies of (**a**, **b**) shortwave (SW) and (**c**, **d**) longwave (LW) budgets
(Units: W m$^{-2}$) with regard to their respective multi-model means for present-day climate under (**a**, **c**) all-
sky and (**b**, **d**) clear-sky conditions over East Asia as simulated by various CMIP6 models. The black, red,
blue, and green lines represent the TOA incoming solar radiation, as well as the net SW/LW radiation at
the TOA, within the atmosphere, and at the surface, respectively.

**Table 1**. Annual land mean estimates (Units: W m$^{-2}$) of the magnitudes of various energy balance
components and cloud radiative effects (CREs) over East Asia under all-sky and clear-sky conditions at the
TOA, within the atmosphere, and at the surface, respectively. The CMIP6 model statistics (e.g., available
model number, spread, standard deviation (SD)), as well as the corresponding multi-model mean, ERA5-,
and CERES-derived estimates are also given in the Table.

| Component (W m$^{-2}$) | CMIP6 | | | | ERA5 | CERES |
|---|---|---|---|---|---|---|
| | models | spread | SD | mean | | |
| **TOA** | | | | | | |
| Solar down | 39 | 4 | 0.2 | 334 | 334 | 334 |
| Solar up all-sky | 39 | 23 | 6 | -117 | -115 | -118 |


| | | | | | | |
|---|---|---|---|---|---|---|
| Solar net all-sky | 39 | 22 | 6.1 | 217 | 219 | 216 |
| Solar up clear-sky | 39 | 24 | 7 | -76 | -78 | -72 |
| Solar net clear-sky | 39 | 24 | 6.9 | 258 | 256 | 262 |
| SW CRE | 39 | 26 | 6.5 | -41 | -37 | -46 |
| Thermal up all-sky | 39 | 12 | 3.5 | -224 | -225 | -226 |
| Thermal up clear-sky | 39 | 15 | 3.2 | -247 | -246 | -250 |
| LW CRE | 39 | 12 | 2.4 | 23 | 21 | 24 |
| Net CRE | 39 | 24 | 5.8 | -18 | -16 | -22 |
| **Atmosphere** | | | | | | |
| SW absorption all-sky | 39 | 19 | 3.8 | 73 | 78 | 74 |
| SW absorption clear-sky | 35 | 19 | 3.8 | 69 | 77 | 71 |
| SW CRE | 32 | 33 | 6.9 | 4 | 2 | 3 |
| LW net all-sky | 39 | 22 | 5.1 | -152 | -150 | -157 |
| LW net clear-sky | 35 | 16 | 3.6 | -151 | -151 | -154 |
| LW CRE | 32 | 14 | 3.3 | -2 | 1 | -3 |
| Net CRE | 32 | 35 | 7.8 | 1 | 2 | 0 |
| **Surface** | | | | | | |
| SW down all-sky | 39 | 33 | 7.6 | 186 | 191 | 178 |
| SW up all-sky | 39 | 24 | 6.5 | -43 | -50 | -36 |
| SW absorbed all-sky | 39 | 23 | 6.1 | 144 | 141 | 142 |
| SW down clear-sky | 35 | 25 | 4.6 | 242 | 238 | 236 |
| SW up clear-sky | 35 | 27 | 6.8 | -53 | -59 | -45 |
| SW absorbed clear-sky | 32 | 36 | 7.8 | 189 | 179 | 191 |
| SW CRE | 35 | 28 | 6.6 | -46 | -38 | -49 |
| LW down all-sky | 39 | 27 | 7.9 | 280 | 273 | 285 |
| LW up all-sky | 39 | 23 | 7.1 | -352 | -347 | -354 |
| LW net all-sky | 39 | 23 | 5.7 | -71 | -74 | -69 |
| LW down clear-sky | 35 | 26 | 6.8 | 256 | 253 | 256 |
| LW up clear-sky | 35 | 23 | 7.1 | -351 | -347 | -353 |
| LW net clear-sky | 35 | 18 | 4.1 | -95 | -94 | -97 |
| LW CRE | 35 | 12 | 3.5 | 24 | 20 | 27 |
| net CRE | 32 | 31 | 6 | -21 | -18 | -22 |
| net radiation | 39 | 20 | 5.3 | 72 | 67 | 73 |
| LH | 40 | 26 | 4.7 | -43 | -38 | — |
| SH | 40 | 21 | 5.2 | -31 | -29 | — |


The simulated SSR, however, shows the largest spread of more than 30 W m$^{-2}$ (ranging from 172–205
W m$^{-2}$) among all the substantially differing all-sky surface radiation components, with a large SD of 7.6 W
m$^{-2}$ (Fig. 2a; Table 1). The multi-model mean SSR is estimated to be 186 W m$^{-2}$, suggesting positive and
negative deviations of 8 and 5 W m$^{-2}$ from the CERES- and ERA5- derived estimates, respectively (Table
1). Interestingly, although the discrepancy between them is very large (8 or 5 W m$^{-2}$), both the resulting
surface solar absorption differences are very small (within 3 W m$^{-2}$), indicating that a higher SSR goes



together with a higher surface albedo (Table 1), which agrees well with that on a global mean level (Wild et
al., 2015).

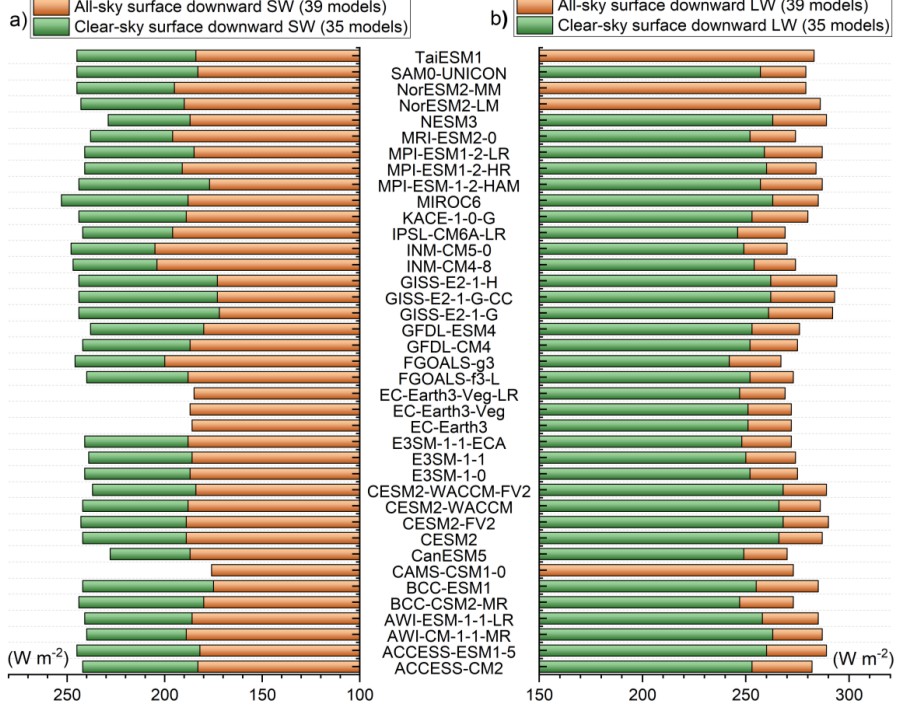

**Figure 2**. Annual land mean surface downward (**a**) SW and (**b**) LW radiation (Units: W m$^{-2}$) under both
all-sky (orange bars) and clear-sky (green bars) conditions over East Asia as calculated by various CMIP6
models.

3.2.    Best estimates for the surface downward SW radiation

As a major component of Earth's energy balance, the solar radiation reaching the Earth's surface

governs a wide range of surface physical and chemical processes. The spatial distributions of the site-based
annual mean SSR from the CMA and GEBA (Section 2.1) over East Asia under all-sky conditions are
presented in Fig. 3a, together with the classified rural and urban sites. In short, the high values are mainly
located at the high elevation stations over western China and a few island sites in Japan (e.g.,
Minamitorishima, Japan; not shown in the figure), especially over the TP, with the largest value reaching
263 W m$^{-2}$ (Geer, Tibet), which is associated with the high atmospheric transparency over these regions.
However, the low annual mean values are primary over southwestern China, with the smallest value of 103
W m$^{-2}$ (Shapingba, Chongqing), which is possibly caused by the higher aerosol loadings (Liao et al., 2015;





de Leeuw et al., 2018) and more clouds (Li et al., 2017; You et al., 2019; Lei et al., 2020; Zhang et al., 2020)
over these regions. This distribution pattern is highly consistent with that over China documented by Wang
et al. (2021).

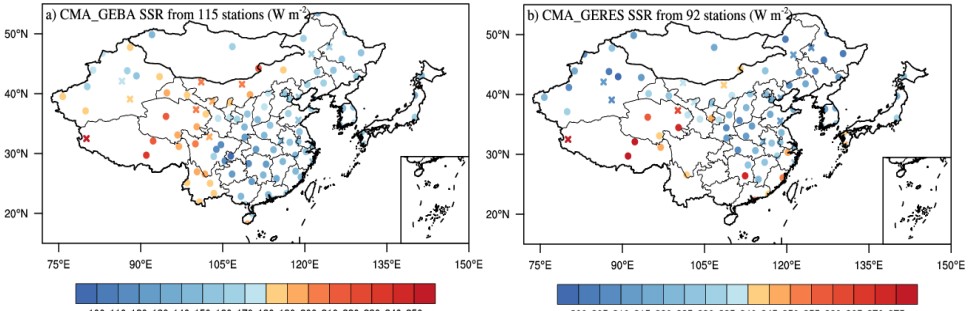

**Figure 3**. Spatial distributions of annual mean surface downward solar radiation (SSR) (Units: W m$^{-2}$) under
(**a**) all-sky and (**b**) clear-sky conditions over East Asia. The all-sky sites are available from 99 CMA (China)
and 16 GEBA (remaining regions outside China and one site in Hongkong, China) stations, while there are
76 CMA and 16 CERES-interpolated sites for clear-sky conditions. The cross and circle symbols indicate
rural (19 vs. 18 for all-sky and clear-sky conditions) and urban stations (96 vs. 74), respectively.

Figure 4 shows the distributions of annual mean SSR biases derived from the CERES, CMIP6 multi-
model mean, and ERA5 against the surface observations, as well as the comparisons of their respective
annual land means at the surface sites with their observed counterparts. The corresponding quantifications
of the magnitudes of station-mean biases are also given in Table 2. According to the comparisons, they all
correlate well with the ground-based observations, with their respective high correlation coefficients of 0.93,
0.87, and 0.89, indicative of the highest accuracy in the CERES-derived estimate (Figs. 4b, d, and f). To
quantify their SSR mean biases against the corresponding observed counterparts, the CERES-based bias at
all sites is the smallest, with a station-mean bias of 3.8 W m$^{-2}$, followed by the CMIP6 multi-model mean
and the ERA5 reanalysis (with respective station-mean biases of 13.8 and 16.5 W m$^{-2}$) (Table 2).
Additionally, among all the aforementioned SSR estimates, the East Asian urban sites are in general more
significantly overestimated than the rural sites on average compared to the surface observations (Figs. 4b, d,
and f; Table 2). This further supports the argument that rural stations might be more representative for larger
scale comparisons (e.g., the general circulation model grid scales) than the urban stations (which are
vulnerable to local pollution) (Wang et al., 2018). The overestimations are mainly located in the high-latitude
regions over East Asia for CERES-derived estimates (among them the underestimations mostly from rural
sites), while the underestimates are primarily located in lower-latitude and eastern coastal regions (Figs. 4a
and b). The CMIP6 multi-model mean and ERA5-derived SSR generally greatly overestimate the surface-
based observations both at urban and rural sites, except for the regions over northern and northeastern Inner
Mongolia, northwestern Heilongjiang (located in the northeastern China), and some individual sites over
southwestern China (Figs. 4c-f). The annual land-mean area-weighted average SSR over East Asia derived



from CERES is estimated to be 178 W m$^{-2}$, which is closest to the surface observational estimate of 174 W
m$^{-2}$, compared to the much higher overestimations of both the CMIP6 multi-model mean and ERA5 (186
and 191W m$^{-2}$) against the surface observations (Table 3), which shows a high consistency with their bias
distributions and the collocated quantifications (Fig. 4; Table 2).

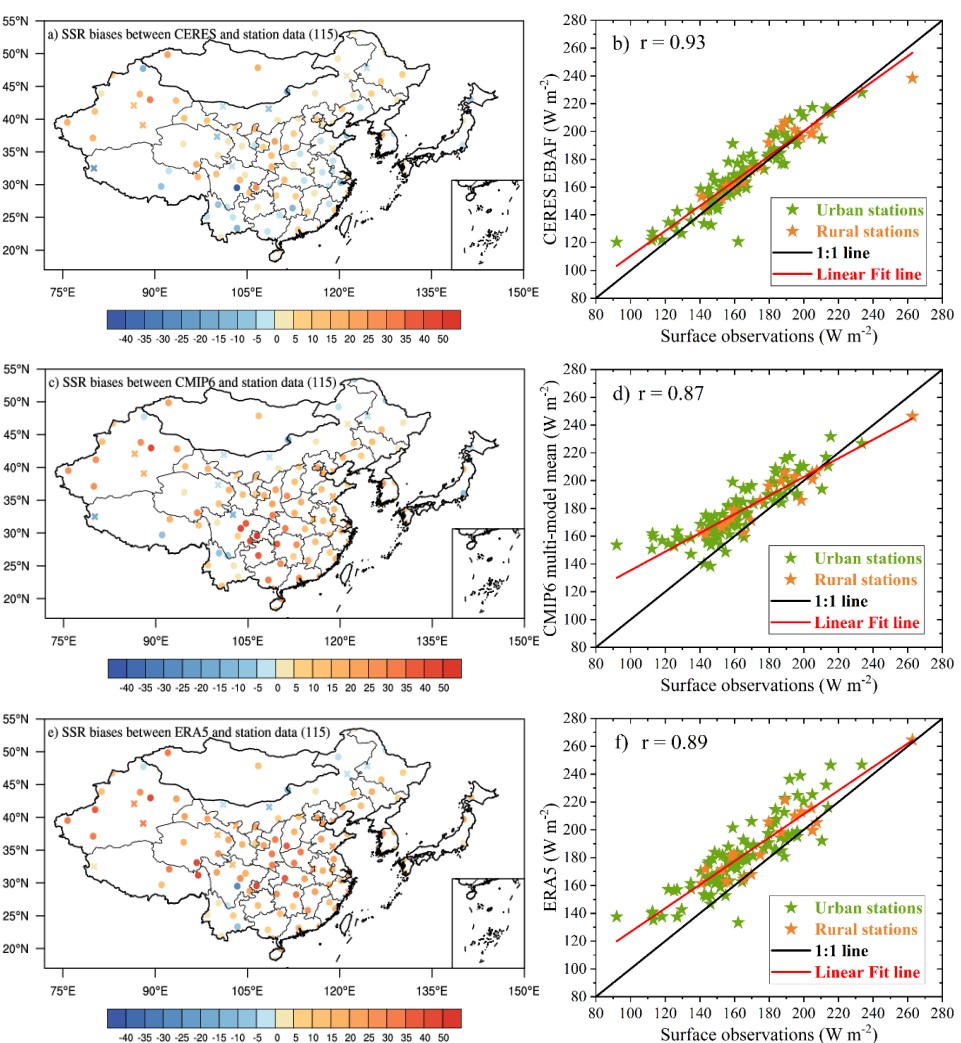

**Figure 4.** Spatial distributions of annual mean SSR biases (Units: W m$^{-2}$) derived from (**a**) CERES-EBAF,
(**c**) CMIP6 multi-model mean, and (**e**) ERA5 reanalysis at a combination of the CMA and GEBA sites under
all-sky conditions over East Asia. The corresponding comparisons of their respective annual means at the
surface sites with their observed counterparts are displayed in (**b**), (**d**), and (**f**), respectively. The cross and
circle symbols in Figs. **a, c, e** as well as the orange and green stars in Figs. **b, d, f** indicate rural and urban
stations, respectively.



**Table 2.** Annual station-mean SSR biases (Units: W m$^{-2}$) derived from CERES-EBAF, CMIP6 multi-model mean, and ERA5 compared to the surface observational sites under all-sky and clear-sky conditions during 2010-2014 over East Asian land, together with the separate station averages of biases over urban and rural sites.

| Station-mean SSR biases | All-sky | | | Clear-sky | | |
|---|---|---|---|---|---|---|
| against surface sites (Units: W m$^{-2}$) | all | urban | rural | all | urban | rural |
| CERES-EBAF - surface sites | 3.8 | 4.2 | 1.6 | 0.4 | 0.5 | -0.3 |
| CMIP6 - surface sites | 13.8 | 15.0 | 7.4 | 9.1 | 9.7 | 6.4 |
| ERA5 - surface sites | 16.5 | 17.2 | 12.7 | 5.7 | 6.2 | 3.6 |

**Table 3.** Annual land mean area-weighted average SSR (Units: W m$^{-2}$) from a combination of the CMA and GEBA (CERES-interpolated) site observations under all-sky (clear-sky) conditions during the period 2010-2014 over East Asia, together with the corresponding estimates from the CERES-EBAF, CMIP6 multi-model means, and ERA5, respectively.

| Average annual mean SSR during 2010-2014 over East Asia (Units: W m$^{-2}$) | Surface observations | CERES-EBAF | CMIP6 | ERA5 |
|---|---|---|---|---|
| All-sky | 174 | 178 | 186 | 191 |
| Clear-sky | 230 | 236 | 242 | 238 |

However, the ground-based observations are spatially limited with sparse stations in some remote regions and are thus inadequate for many applications, as they may be not representative for real situations. To better constrain the large spread in the model-based SSR outlined above, we combine the ground-based observations to obtain the best estimate referring to the approach introduced in (Wild et al., 2013a). Figure 5a gives various CMIP6 model biases of all-sky SSR at all the surface sites and their respective East Asian land means. The higher overestimations relative to surface observations generally correspond to higher model-based East Asian land means, with a much higher correlation coefficient of 0.96 than that of 0.88 on the global scale (Wild et al., 2015). Thus, the best estimate of the annual East Asian land-mean SSR is deduced to be 174.2±1.3 W m$^{-2}$ ($2\sigma$ uncertainty) in light of the linear regression analysis. The corresponding estimates from CERES and ERA5 are also labeled in the figure, at 178 and 191 W m$^{-2}$, respectively, implying a slight and substantial overestimation for CERES and ERA5 estimates. There is an overall tendency that most models overestimate the surface downward SW fluxes (36 out of 39 sites) compared to the ground-based observations, with a multi-model mean overestimation relative to site observations of 13.8 W m$^{-2}$, which is also a longstanding issue in climate modelling (Wild et al., 1995; Wild et al., 2015).

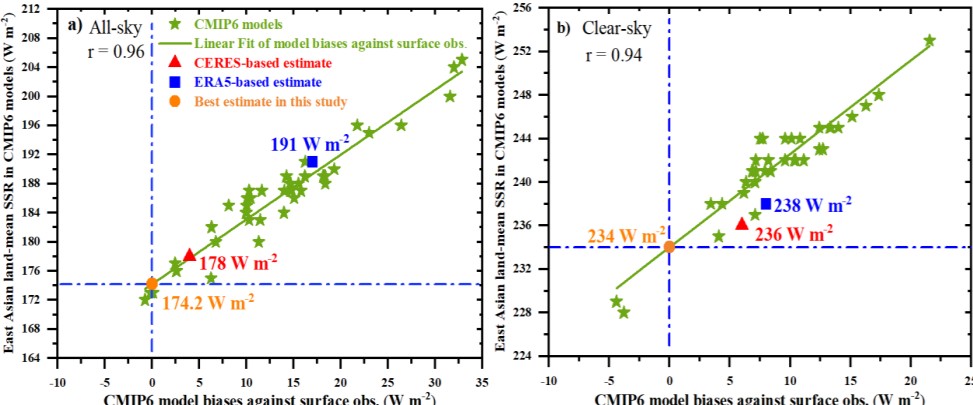

**Figure 5.** Annual land mean SSR (Units: W m$^{-2}$) of various CMIP6 models as well as their respective model biases relative to an average over surface sites (99 CMA and 16 GEBA for all-sky; 76 CMA and 16 CERES-interpolated sites for clear-sky) under (**a**) all-sky and (**b**) clear-sky conditions during 2010-2014 over East Asia. Green stars represent various CMIP6 models. Best estimate here (orange circle) can be inferred from the intersection between the linear regression line (green solid lines) and the zero-bias line (blue dotted lines). Furthermore, the corresponding estimates from CERES-EBAF and ERA5 are also given by red triangle and blue square, respectively.

### 3.3. Longwave components

Similar to the all-sky SW counterparts, obvious discrepancies can still be noted in the annual land-mean LW radiation over East Asia among models, especially for those within the atmosphere and at the surface (Fig. 1c). Correspondingly, the simulated TOA OLR varies in a range of 12 W m$^{-2}$, which is almost 10 W m$^{-2}$ lower than that within the atmosphere (22 W m$^{-2}$) and at the surface (23 W m$^{-2}$) (Table 1). The estimated annual East Asian land-mean TOA OLR from the CMIP6 multi-model mean is -224 W m$^{-2}$, within 2 W m$^{-2}$ of the deviations from the CERES- and ERA5-inferred estimates. The model spread of the simulated annual land-mean net LW radiation becomes larger from the TOA to the surface, with SDs of 3.5, 5.1, and 5.7 W m$^{-2}$, respectively, which shows the same tendency as the relative (percentages) differences with respect to their multi-model means (5.4%, 14.5%, and 32.4%).

These large discrepancies in surface net LW radiation between models are particularly evident in the surface downward LW radiation (Fig. 2b; Table 1), with a range up to 27 W m$^{-2}$ (from 267 to 294 W m$^{-2}$) and a SD of 7.9 W m$^{-2}$, which is also the largest deviation among all components under all-sky conditions. Compared to the CERES estimates, the slightly lower surface upward LW radiation (-352 vs. -354 W m$^{-2}$) and much lower surface downward LW radiation (280 vs. 285 W m$^{-2}$) from the multi-model means are the major reason for the small deviation (within 2 W m$^{-2}$) of the surface net LW radiation between them (Table 1). It's interesting to note that the annual East Asian land-mean surface upward LW radiation estimated from the ERA5 is the lowest among all these estimates, at -347 W m$^{-2}$, suggesting the lowest surface skin temperature of the ERA5 product according to the Stefan-Boltzmann law, followed by the estimates from the multi-model mean and CERES (Table 1). In addition, the annual land-mean surface downward LW radiation estimated by ERA5 is 273 W m$^{-2}$, approximately 7 and 12 W m$^{-2}$ lower than the estimates by the





CMIP6 multi-model mean and CERES, respectively (Table 1). Therefore, both the lower surface upward
and downward LW radiation fluxes result in the small deviation in the estimated surface net LW radiation
from ERA5 compared to those from the multi-model mean and CERES (Table 1). Since the reanalysis
products take as many observed atmospheric parameters with global coverage as possible into consideration
during the radiative transfer calculations, they are widely used to obtain more accurate surface LW radiation
(Simmons et al., 2004; Wild et al., 2015). We also examined the corresponding surface LW fluxes from
another reanalysis, namely MERRA-2, and found much lower annual land means than those from ERA5, in
particular for the surface downward LW radiation (not shown), which arrives at the similar conclusions with
that documented by Urraca et al. (2018). Thus, considering the limited observational surface LW radiation
data over East Asia, ERA5 might be the best reference for the estimates of the annual land-mean surface
upward and downward LW radiation, at -347 and 273 W m$^{-2}$, respectively (Table 1).

### 3.4.    Discussion of land energy balance over East Asia under all-sky conditions
#### 3.4.1.    Radiative components
Figure 6a displays the schematic diagram of the all-sky land mean energy balance over East Asia,
including the above-mentioned SW and LW radiation budgets and other radiative components discussed in
the following. The estimated annual East Asian land-mean incoming, reflected, and net SW radiation as well
as the OLR at the TOA are therefore 334, -118, 216, and -226 W m$^{-2}$ (Table 1), respectively, based on the
CERES EBAF dataset. The corresponding uncertainties are obtained from the uncertainty of 2.5 ($1\sigma$
uncertainty) W m$^{-2}$ for both SW and LW fluxes given by (Loeb et al., 2018). The annual East Asian land-
mean TOA OLR in CERES-EBAF is estimated to be 10 W m$^{-2}$ larger than the TOA absorbed SW radiation,
implying an energy loss of 10 W m$^{-2}$ at the TOA under all-sky conditions, which should be compensated by
the LH and SH transported from regions outside East Asia (Fig. 6a).






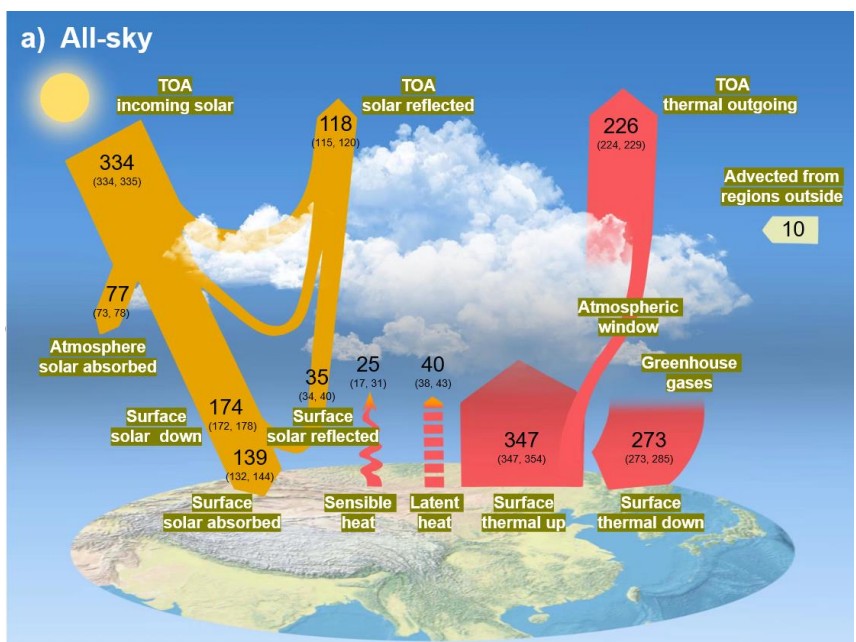


**Figure 6.** Diagrams of the annual land mean energy balance (Units: W m⁻²) over East Asia under (**a**) all-sky and (**b**) clear-sky conditions for present-day climate. The uncertainty ranges are also given in parentheses.






For the SSR, the annual East Asian land-mean best estimate based on the CMIP6 multi-model

simulations and surface observations is 174.2 W m$^{-2}$ (Fig. 5a and Fig. 6a). Considering the abnormally high
overestimation by ERA5 compared to surface observation, the high value of the uncertainty range is given
by the estimate from CERES EBAF (178 W m$^{-2}$), while its low value is from the lowest model estimate (172
W m$^{-2}$; Fig. 2a) (Fig. 6a). The all-sky surface albedo information is derived from the ratio between the
CERES-derived surface upward and downward solar radiation, with a radiation weighted average of around
0.2 (36.4/178.3) over East Asian land. However, the corresponding surface albedos estimated by the CMIP6
multi-model mean and ERA5 are substantially higher than that from the CERES, with respective averages
of around 0.23 (42.7/186.4) and 0.26 (49.6/191). Considering the large spatial coverage of remote sensing
measurement to map albedo globally, the CERES-derived annual East Asian land-mean surface albedo is
adopted as the best estimate in this study. Therefore, considering the rounded best SSR estimate of 174 W
m$^{-2}$, the calculated surface reflected and absorbed SW radiation fluxes are around -35 and 139 W m$^{-2}$,
respectively. As shown in Table 1, the uncertainty range of the surface absorbed SW radiation is 132–144
W m$^{-2}$ according to the lowest value of CMIP6 models and the highest estimate among the aforementioned
estimates, which gives rise to an uncertainty range of the surface reflected solar radiation of 34–40 W m$^{-2}$.
Together with the annual East Asian land-mean SW absorption at the TOA and surface of 216 and 139 W
m$^{-2}$, the best estimate for the atmospheric SW absorption is therefore to be 77 W m$^{-2}$, which is within 4 W
m$^{-2}$ of the differences between those estimated from the CMIP6 multi-model mean and CERES and closes
to the ERA5-derived estimate of 78 W m$^{-2}$ (Table 1). The uncertainty range of the atmospheric SW
absorption is also determined by the estimates from different data sources as shown in Fig. 6a.

The downward LW radiation emitted by the atmosphere is mainly sensitive to the near-surface

temperature, water vapor, and cloud properties, while the surface emission is in proportion to the skin
temperature according to the Stefan-Boltzmann law. As analyzed in section 3.3, the best estimates of the
East Asian annual land-mean surface upward and downward LW radiation amount to -347 and 273 W m$^{-2}$,
respectively, with uncertainty ranges coming also from the above-discussed different data sources (Fig. 6a).
The surface net LW radiation is then estimated to be -74 W m$^{-2}$ based on the surface upward and downward
LW radiation outlined above. Combined with TOA outgoing thermal radiation of -226 W m$^{-2}$, the estimated
atmospheric net LW radiation is -152 W m$^{-2}$, which is close to the collocated estimates from the multi-model
mean (-152 W m$^{-2}$) and ERA5 (-150 W m$^{-2}$) but deviates substantially from the CERES-derived estimate of
-157 W m$^{-2}$ (Table 1). Considering the surface absorbed SW radiation of 139 W m$^{-2}$, a best estimate for
surface net radiation is 65 W m$^{-2}$, suggesting that around 65 W m$^{-2}$ of energy is available for the non-radiative
SH and LH. Besides, the ERA5 estimate of 67 W m$^{-2}$ is very close to the best estimate of 65 W m$^{-2}$, while
much higher estimates of 72 and 73 W m$^{-2}$ are obtained from the multi-model mean and CERES (Table 1),
respectively.



### 3.4.2. Nonradiative components

The surface net radiation is mainly balanced by the non-radiative components of SH and LH in addition to a very small proportion of ground heat flux and melt (less than 1%) (Ohmura, 2004). However, due to the lack of constraints from in-situ and space observations, this partitioning of the surface net radiation into SH and LH is still subject to considerable uncertainties. As shown in Fig. S2, the simulated annual East Asian land-mean LH and SH vary greatly between different models, with a range of 26 and 21 W m$^{-2}$, respectively, as well as the relative discrepancies relative to their respective multi-model means of 60% ($\frac{26}{43}\times100\%$) and 68% ($\frac{21}{31}\times100\%$), respectively, showing larger discrepancies between models with larger uncertainties in SH (Table 1). The best SH estimate can therefore be obtained from the residual of the LH. To obtain a more accurate surface LH from available datasets of the multi-model mean and ERA5, we take an average of them as the best estimate, namely -40 W m$^{-2}$, the uncertainty ranges of which are also given according to these estimates (Fig. 6a). Note that all the values in this study are calculated on the basis of one decimal point, which may result in 1 W m$^{-2}$ of bias during the rounding process. Combined with the surface net radiation and LH of 65 and -40 W m$^{-2}$, respectively, the surface SH is estimated to be -25 W m$^{-2}$, the uncertainty range of which is also given by the existing estimates from various CMIP6 models and ERA5 (Fig. 6a). In addition, although the annual land-mean SH estimated from the MERRA-2 is much higher than the estimates from multi-model mean and ERA5 (not shown), the estimated LH is around -39 W m$^{-2}$ (not shown), very close to the best estimate of -40 W m$^{-2}$, which increases our confidence in the estimation of this quantity.

### 3.4.3. Comparisons with global annual land-mean estimates

Notable discrepancies exist in the global land-mean energy budgets reported by Wild et al. (2015) and the regional land-mean estimates over East Asia in this study (Fig. S3; Table 4). For the SW budgets, the estimated annual land-mean TOA incident solar radiation over East Asia is 9 W m$^{-2}$ higher than that over global land (334 vs. 325 W m$^{-2}$), implying a slightly lower land-mean solar zenith angle over East Asia. Comparisons also show a slightly higher relative percentage of TOA reflected solar radiation of 0.8% despite of the much lower surface reflected SW radiation of 4.3% over East Asian land compared to global land with respect to their respective TOA incident solar radiation (thereafter call 'relative percentage' for short). This suggests much more relative atmospheric SW reflection of 5.2% over East Asian land, which agrees fairly well with more aerosols (Wei et al., 2019) and clouds (King et al., 2013; Fan et al., 2018; also see Fig. S4) over this region compared to global land. However, the annual land-mean solar radiation reaching the East Asian surface is around 10 W m$^{-2}$ lower than that over global land (174 vs. 184 W m$^{-2}$), approximately accounting for 52.1% and 56.6% of their respective incident solar radiation at the TOA, respectively, indicating lower fraction of solar energy arriving at the East Asian surface compared to global land. Together with the lower annual land-mean surface albedo over East Asian land compared to global land (20% vs. 26%), this leads to the similar relative percentages of surface absorptions (41.6% vs. 41.9%). Although the magnitude of the atmospheric SW absorptions over East Asian and global land are nearly the same (both around 77 W m$^{-2}$), the corresponding relative percentage over East Asian land is a little bit lower than that





over global land (around 0.6%). This is somewhat unexpected due to the fact of more clouds and aerosol
loadings over East Asian land, which is possibly offset by the lower water vapor contents caused by the
higher altitudes and thinner air over the TP.

**Table 4.** Comparisons of the annual mean SW/LW energy balance components (Units: W m$^{-2}$) over East
Asian land (this study) and global land (Wild et al., 2015) as well as the corresponding relative percentages
with regard to their respective TOA incident solar radiation/surface LW emissions, along with the relative
percentage differences between them.

| Component | East Asian land | | Global land | | Percentage difference |
|---|---|---|---|---|---|
| | Annual mean | Relative percentage | Annual mean | Relative percentage | |
| **SW budget** | | | | | |
| TOA solar down | 334 | 1 | 325 | 1 | — |
| TOA solar up | -118 | 35.3% | -112 | 34.5% | 0.8% |
| Atmospheric SW absorption | 77 | 23.1% | 77 | 23.7% | -0.6% |
| Atmospheric SW reflection | -83 | 24.9% | -64 | 19.7% | 5.2% |
| Surface solar down | 174 | 52.1% | 184 | 56.6% | -4.5% |
| Surface solar up | -35 | 10.5% | -48 | 14.8% | -4.3% |
| Surface solar absorption | 139 | 41.6% | 136 | 41.9% | -0.3% |
| **LW budget** | | | | | |
| TOA LW up | -226 | 65.1% | -232 | 62.4% | 2.7% |
| Atmospheric LW absorption | -152 | 43.8% | -166 | 44.6% | -0.8% |
| surface LW down | 273 | 78.7% | 306 | 82.3% | -3.6% |
| Surface LW up | -347 | 1 | -372 | 1 | — |


For the LW budgets, the regional surface LW emission over East Asia is estimated to be much lower
than the global land-mean estimates in Wild et al. (2015) (Fig. S3), which mainly results from the lower
temperature over the TP induced by high altitudes. The relative percentage of land mean surface downward
LW radiation with respect to the surface emission over East Asia is about 78.7 %, which is lower than the
global estimate of 82.3%, corresponding well to a reduction in greenhouse effect and fewer low clouds due
to the TP (Fig. S4) considering its coverage over East Asian land. Ultimately, a higher percentage of LW
radiation is emitted to space over East Asian land compared to global land (65.1% vs. 62.4%). Our estimates
also indicate approximately similar amounts of LH (40 vs. 38 W m$^{-2}$) and much lower SH (25 vs. 32 W m$^{-2}$
$^{2}$) over East Asia compared to the global land-mean estimates (Fig. S3), which is possibly related to the
lower East Asian-land surface temperature.
In general, as can be concluded from Table 4, although much less surface SW radiation of 4.3% is
reflected over East Asian land compared to global land, a slightly more SW reflection of 0.8% is estimated
at the TOA, indicating much larger atmospheric SW reflection of 5.2% due to the stronger scattering from
aerosols and clouds over East Asian land than global land. However, the SW absorption within the
atmosphere over East Asian land is 0.6% lower than that over global land despite of the more absorption
from clouds and aerosols, which is possibly offset by the lower water vapor contents caused by the thinner
air over the TP. The lower surface temperature, weaker greenhouse effect and fewer low clouds due to the
high altitudes and the thinner air over the TP in East Asian land are the major reasons for the relative lower

On





surface LW emission, less and more fractions of surface downward LW radiation of 3.6% and the OLR of
2.7% over East Asian land compared to global land, respectively.

## 4.   Assessment of land energy balance budgets under clear-sky conditions

The clear-sky land energy balance budgets over East Asia are similarly evaluated as all-sky conditions.
Detailed analyses are given in Supplemental material if interested. The annual land-mean SW clear-sky
absorptions at the TOA and surface over East Asia show larger variations among different models than that
under all-sky conditions (Fig. 1a and b; Table 1), which is consistent with that reported by Wild et al. (2019)
but is amazingly in contrast to the recognition that the representation of clouds is the largest uncertainties in
climate models (Dolinar et al., 2015). Specially, the surface SW clear-sky absorptions simulated by various
models still exhibit a larger uncertainty than the TOA counterparts despite of the lower absolute values (Fig.
1b; Table 1). Contrary to the all-sky counterparts, the simulated clear-sky SSR among different models,
shows notably smaller inter-model spread and SD than the surface SW absorptions (Table 1), with much
smaller model discrepancy compared to the all-sky conditions (Fig. 2a; Table 1).
To further constrain the outlined inter-model discrepancy of the simulated clear-sky SSR, surface
observations from the CMA and CERES-interpolated estimates at the GEBA sites are utilized in this study.
The high values of the station-based clear-sky SSR are mainly located in the TP, but with an abnormally
high value located at the southern China (Fig. 3b). All the East Asian land-mean clear-sky SSR estimates
from CERES, CMIP6 multi-model mean, and ERA5 agree reasonably well with the surface observations,
but with smaller correlation coefficients ranging from 0.78 to 0.82 compared to the all-sky conditions (Figs.
7 b, d, and f). The CERES-derived clear-sky SSR is mainly overestimated in central and western China, but
with slight underestimations mainly located in northeastern, eastern, and southern China (Fig. 7a). Similar
bias patterns can also be found in the clear-sky SSR from the CMIP6 multi-model mean and ERA5 compared
to the surface observations, except for some individual sites over northeastern Inner Mongolia, eastern China,
western Mongolia, and Japan (Figs. 7c and e), but with relatively smaller overestimations than the all-sky
counterparts (Figs. 4c and e; Table 2). Specifically, the smallest station mean bias in CERES-derived SSR
compared to the multi-model mean and ERA5 (Table 2) can be attributed to its even distributed surface sites
of overestimations and underestimations (Figs. 7b, d, f). Again, among all the aforementioned clear-sky SSR
biases, more overestimations exist in urban stations than the rural stations (b, d, f in Figs. 4 and 7; Table 2).
Consequently, all East Asian land-mean area-weighted averages of clear-sky SSR from CERES, CMIP6
multi-model mean, and ERA5 show higher overestimations of around 6, 12, and 8 W m$^{-2}$, respectively,
compared to the surface observed counterpart of 230 W m$^{-2}$ (Table 3). Based on the similar method
introduced in Wild et al. (2015), the best estimate for the East Asian land-mean clear-sky SSR is determined
to be 234±1.1 W m$^{-2}$ (2$\sigma$ uncertainty), with a slightly smaller correlation coefficient of 0.94 and smaller
deviations from the CERES and ERA5 estimates compared to the all-sky counterparts (Fig. 5b; Table 3).
Besides, the overestimations still exist in the observed land-mean clear-sky SSR for most climate models
over East Asia, with a smaller multi-model mean overestimation of 9.1 W m$^{-2}$ than the all-sky counterparts.

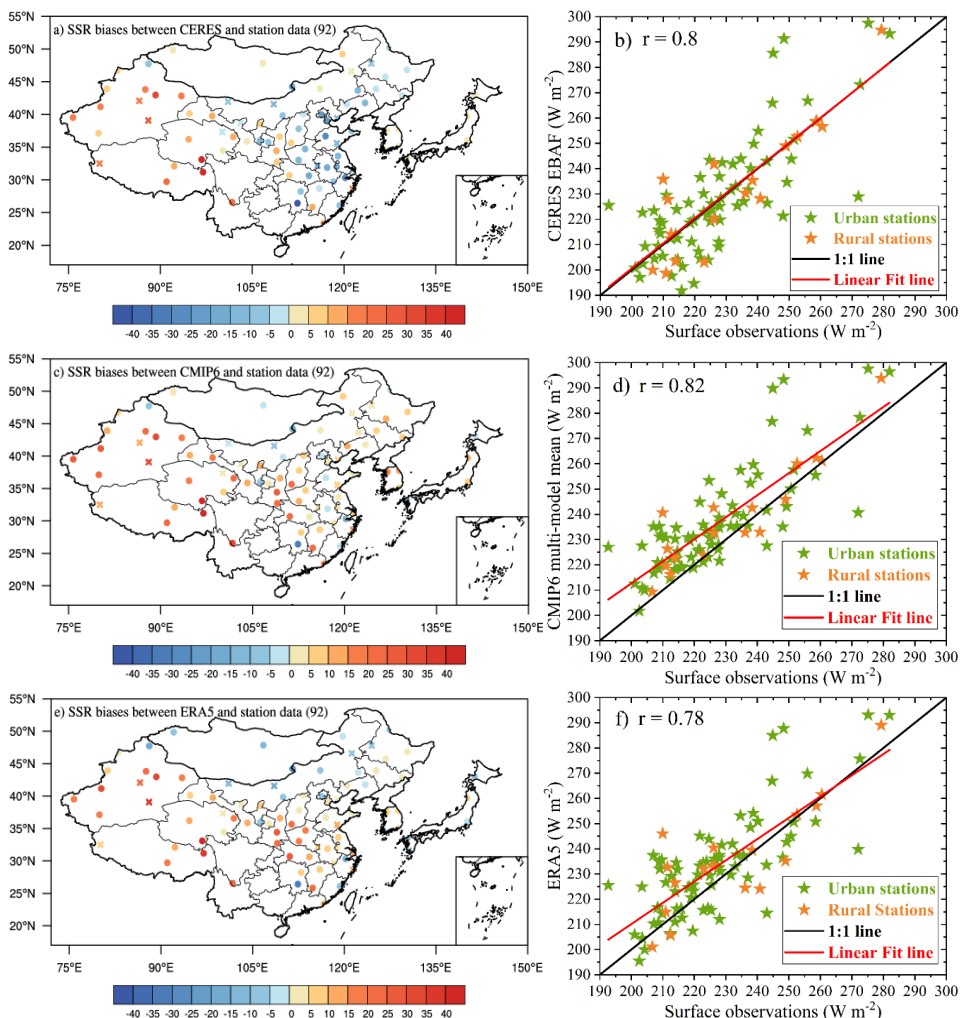


**Figure 7.** Spatial distributions of annual mean SSR biases derived from (**a**) CERES-EBAF, (**b**) CMIP6 multi-model mean, and (**c**) ERA5 reanalysis against surface observations from a combination of the CMA and CERES-interpolated sites under clear-sky conditions over East Asia. The corresponding comparisons of their respective annual land means at the surface sites with their observed counterparts are displayed in (**b**), (**d**), and (**f**), respectively. The cross and circle symbols in Figs. **a, c, e** as well as the orange and green stars in Figs. **b, d, f** indicate rural and urban stations, respectively.


This clear-sky energy budget only represents the removal of cloud but maintains the same atmospheric

conditions as the all-sky conditions. Ultimately, the clear-sky East Asian land-mean energy budget is
established as displayed in Fig. 6b. In addition to the analyses above, the clear-sky TOA energy budgets are
derived from CERES-derived product, with uncertainty ranges referred to Loeb et al. (2018), while the
surface LW budgets are again from ERA5 reanalysis. Also, additional clear-sky radiation weighted surface



albedo of 0.19 from CERES is obtained to estimate the surface reflected and absorbed SW radiation. All the
uncertainty ranges are given by different data sources from various CMIP6 models, as well as the multi-
model mean, CERES-, and ERA5-derived estimates, except for their TOA counterparts.

We doublecheck the energy balance components evaluated in this study by referring to the uncertainty

ranges from CERES-derived product given by Kato et al. (2018) (Table 5), which indicates that all estimated
energy components fall within these uncertainty ranges, except for the all-sky surface downward LW
radiation, with about 3 W m$^{-2}$ lower than the corresponding lowest CERES range. This is in line with its
much higher CERES-derived estimate compared to that of the ERA5 (285 vs. 273 W m$^{-2}$) (Table 1).

**Table 5.** Uncertainties (Units: W m$^{-2}$) in 1°×1° regional monthly surface SW, LW, and net (SW + LW)
fluxes under all-sky and clear-sky conditions for the CERES-EBAF Edition 4.1 product (referring to Kato
et al. (2018)), as well as its corresponding estimates of various surface fluxes.

| Uncertainties($1\sigma$) | All-sky | Clear-sky |
|---|---|---|
| SW down | 178±14 | 236±6 |
| SW up | 36±11 | 45±11 |
| SW net | 142±13 | 191±13 |
| LW down | 285±9 | 256±8 |
| LW up | 354±15 | 353±15 |
| LW net | 69±17 | 97±17 |
| SW + LW net | 73±20 | 95±20 |


Overall, around 21.6% and 56.9% of the TOA incoming solar radiation are absorbed by the atmosphere

and surface, respectively, for clear-sky conditions, while these absorptions are 23.1% and 41.6% for all-sky
conditions. This implies that the existence of clouds results in more atmospheric SW absorption of around
1.5% and much less surface solar absorption of around 15.3% with respect to the TOA incoming solar
radiation.

**5.    The cloud radiative effects (CREs)**

According to the annual land-mean best estimates of radiative components over East Asia under all-sky

and clear-sky conditions obtained in previous sections, the present-day CREs can be inferred quantitively
over this region. The calculated SW, LW, and net CREs at the TOA, within the atmosphere, and at the
surface are therefore presented in Fig. 8. Moreover, the corresponding calculation formulas are also given in
the followings:

TOA SW CRE  = TOA outgoing SW$_{\text{all-sky}}$- TOA outgoing SW$_{\text{clear-sky}}$

TOA LW CRE  = TOA outgoing LW$_{\text{all-sky}}$- TOA outgoing LW$_{\text{clear-sky}}$

533        TOA Net CRE  = TOA SW CRE  +  TOA LW CRE




$$\text{Surface Net SW CRE} = \text{Surface Net SW}_{\text{all-sky}} - \text{Surface Net SW}_{\text{clear-sky}}$$


$$\text{Surface Net LW CRE} = \text{Surface Net LW}_{\text{all-sky}} - \text{Surface Net LW}_{\text{clear-sky}}$$


$$\text{Surface Net total CRE} = \text{Surface Net SW CRE} + \text{Surface Net LW CRE}$$



$$\text{Atmospheric SW CRE} = \text{TOA SW CRE} - \text{Surface Net SW CRE}$$


$$\text{Atmospheric LW CRE} = \text{TOA LW CRE} - \text{Surface Net LW CRE}$$



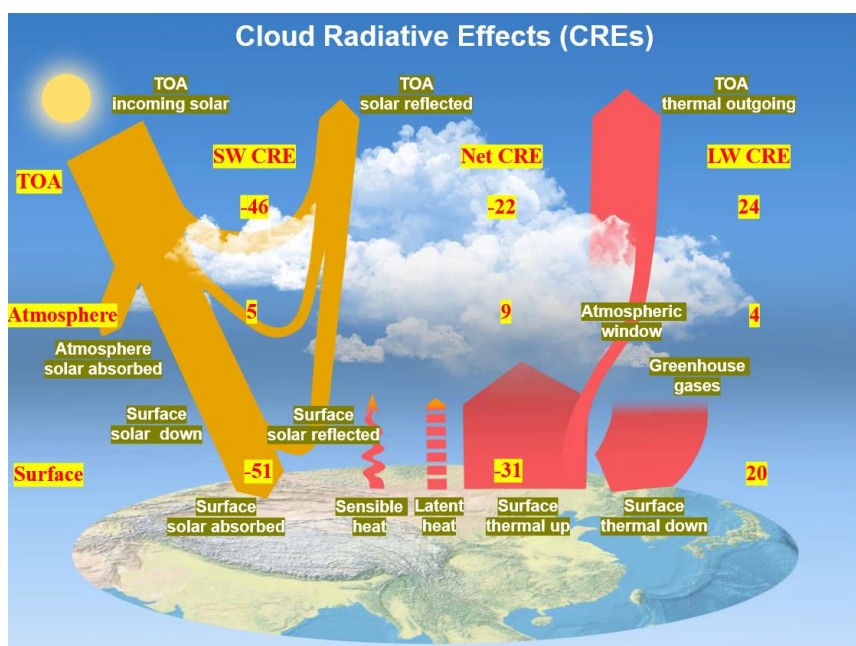


**Figure 8.** Diagram of the annual land mean SW, LW, and net (SW + LW) cloud radiative effects (CREs)
(Units: W m$^{-2}$) at the TOA, within the atmosphere, and at the surface over East Asia, calculated by the
differences between all-sky and clear-sky radiation budgets as given in Fig. 7.


Best estimates for the annual East Asian land-mean reflected solar radiation at the TOA under all-sky
and clear-sky conditions are -118 and -72 W m$^{-2}$, respectively, differing by -46 W m$^{-2}$, indicating that the
clouds give rise to an extra 46 W m$^{-2}$ solar reflection at the TOA, thus cooling the Earth-atmosphere system.
Similarly, the TOA LW CRE, obtained as the difference between the TOA thermal radiation under all-sky
and clear-sky conditions, is 24 W m$^{-2}$, suggesting a warming effect of clouds on the system. Thus, the



estimated TOA net CRE is -22 W m$^{-2}$, pointing out that the overall effects of clouds result in an energy loss
and net cooling to the system, not only in the global mean, but also over East Asian land.
At the Earth's surface, the shading effects of clouds are estimated to reduce the surface solar radiation
by 60 W m$^{-2}$, from 234 to 174 W m$^{-2}$, while the surface solar absorption differs by 51 W m$^{-2}$, from 190 to
139 W m$^{-2}$, namely the surface net SW CRE is -51 W m$^{-2}$. On cloudy skies, the estimated surface downward
LW radiation increases from 253 to 273 W m$^{-2}$, with an increase of 20 W m$^{-2}$, illustrating that the surface
net LW CRE is 20 W m$^{-2}$ and therefore leads to a surface warming. Thus, the surface net CRE, i.e., the sum
of the surface net SW and LW CRE, is then -31 W m$^{-2}$, indicating that clouds contribute more to the SW
energy budgets. Eventually, the clouds lead to the enhancement of the SW and LW absorption within the
atmosphere of around 5 and 4 W m$^{-2}$, respectively, thus resulting in an atmospheric net CRE of 9 W m$^{-2}$ over
East Asian land.
The above CRE estimates are compared to the corresponding estimates from different data sources (Fig.
9; Table 1). Generally, compared to the LW CREs (Fig. 9b), the simulated SW CREs show larger spreads
and SDs amongst models (Fig. 9a; Table 1). For the SW CREs at the TOA, within the atmosphere, and at
the surface, the CERES-derived estimates match perfectly with the best estimates mentioned above, within
2 W m$^{-2}$ of the biases, followed by the estimates from the multi-model means and ERA5 (Table 1). For the
LW CREs, the calculated TOA LW CREs from the CMIP6 multi-model mean and CERES differ by no more
than 1 W m$^{-2}$ compared to the best estimate, while large differences are noted at the surface LW CREs,
thereby leading to their opposite signs in the atmospheric LW CREs (Fig. 9b; Table 1). Specifically, since
the ERA5-based TOA LW CRE deviates by no more than 3 W m$^{-2}$ with the best estimate of 24 W m$^{-2}$ with
nearly the same surface LW CRE, the estimated atmospheric LW CRE is therefore the closest to the best
estimate (Table 1). This is owing to the fact that we make use of the ERA5 data as the reference to estimate
the surface LW radiation. Thus, the major reason for the large discrepancies in the atmospheric and surface
LW CREs estimated from different data sources with respect to the best estimates in this study is the
determination of the surface downward and upward LW radiation, which is also the reason for the large
deviations in their net CREs (Fig. 9c).



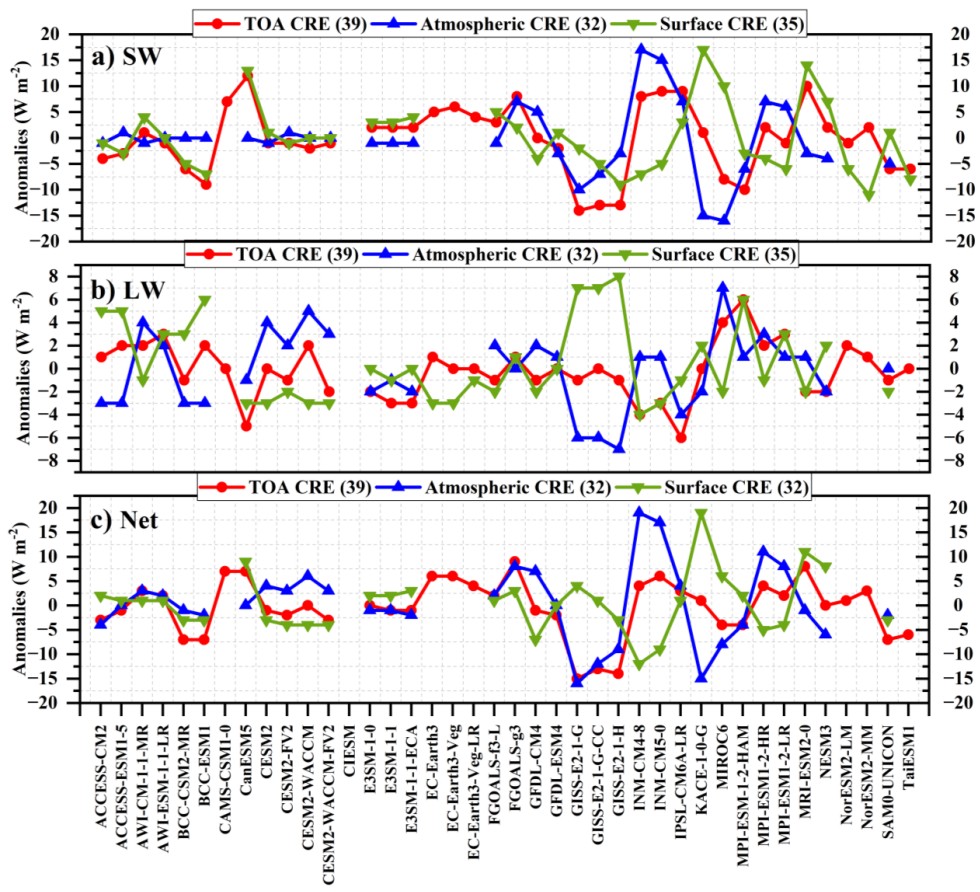

**Figure 9.** Annual land mean anomalies of (**a**) SW, (**b**) LW, and (**c**) net (SW + LW) CRE (Units: W m$^{-2}$) at
the TOA (red line), within the atmosphere (blue line), and at the surface (green line) with regard to their
respective multi-model means over East Asia, respectively, as represented by various CMIP6 models. The
numbers in the parentheses indicate the available CMIP6 climate models for the corresponding radiation
components.

A better comparison with the global annual mean best estimates of CREs by Wild et al. (2019) is given
in Fig. S5. At the TOA, a slightly lower and much lower East Asian land-mean SW and LW CREs of 1 W
m$^{-2}$ and 4 W m$^{-2}$ result in 3 W m$^{-2}$ more energy loss at the TOA compared to the globe. At the surface, much
lower annual East Asian land-mean SW and LW CREs by 3 W m$^{-2}$ and 8 W m$^{-2}$ are estimated compared to
the values over the globe, leading to a net CRE deviation of 5 W m$^{-2}$, indicative of 5 W m$^{-2}$ more energy loss
at the surface. However, lower and higher annual East Asian land-mean SW and LW CREs of 2 and 4 W m$^{-2}$
within the atmosphere contribute to the nearly close net CRE with a deviation of no more than 2 W m$^{-2}$
compared to the global mean estimates. On the whole, lower annual East Asian land-mean best estimates in
the absolute values of surface SW and LW CREs as well as the TOA LW CRE compared to their global
mean counterparts give rise to the CRE differences between them.


## 6. Summary and conclusions

This study aims to explore how the energy budgets are interrupted by the complex orographic and thermal effects of the TP, as well as the high anthropogenic aerosol emissions over East Asian land compared to global land, based on complementary data sources from space and surface observations, as well as the CMIP6 climate models and ERA5 reanalysis. A further quantitative investigation of CREs at the TOA, within the atmosphere, and at the surface is also conducted.

Comparisons between all-sky and clear-sky energy budgets indicate that the overall effects of clouds greatly reduce the surface solar absorption by about 15.3% and enhance that within the atmosphere by 1.5%. Compared to the global land energy budget estimates from Wild et al. (2015), for the SW budgets, notably more atmospheric SW reflection of 5.2% but with a slightly less atmospheric SW absorption of 0.6% with respect to their respective TOA incident solar radiation are estimated over East Asian land, possibly indicating that the lower water vapor content effects due to TP overcompensate for the aerosol and cloud effects over East Asian land. For the LW budgets, a substantially lower surface LW emission of around 25 W m$^{-2}$ and smaller relative surface downward LW radiation of around 3.6% with respect to their respective surface emissions can be noticed over East Asian land compared to global land, which possibly result from the lower regional surface skin temperature, as well as the weaker greenhouse effect and fewer low clouds mainly induced by the high altitude and thinner air over TP, thus leading to a higher percentage of regional OLR of 2.7%.

The CREs over East Asian land are inferred through the energy budget differences between all-sky and clear-sky conditions. The clouds reduce the solar absorption at the TOA by 46 W m$^{-2}$ and enhance the TOA thermal radiation by 24 W m$^{-2}$, respectively, leading to a TOA net CRE of -22 W m$^{-2}$, a more cooling effect on the regional climate system than that over globe (-19 W m$^{-2}$). At the surface, the net CRE is estimated to be -31 W m$^{-2}$ according to less solar absorption of 51 W m$^{-2}$ and more downward thermal radiation of 20 W m$^{-2}$, indicative of larger cloud impacts on SW radiation. Within the atmosphere, the estimated net CRE is 9 W m$^{-2}$ due to an increase of 5 W m$^{-2}$ of solar absorption and 4 W m$^{-2}$ of the net thermal radiation, respectively. Compared to the global mean best estimates of CREs as introduced by Wild et al. (2019), relatively lower East Asian land-mean best estimates of surface SW and LW CREs as well as the TOA LW CRE contribute to the CRE differences between them.

On the whole, all the estimated land-mean energy balance components over East Asia in this study fall within the uncertainty ranges of the CERES-derived assessments, except for the all-sky surface downward LW radiation. More accurate and reliable datasets should be utilized to reduce the substantial uncertainties in the regional energy balance estimates, particularly in the surface budgets, and more widespread temporal and spatial representations of energy budget research are recommended for more comprehensive comparisons in future.





*Acknowledgments.* This research was funded by the National Key Research and Development Program of
China (2017YFA0603502) and the Science and Technology Development Fund of CAMS
(2021KJ004&2022KJ019). The Global Energy Balance Archive (GEBA) is co-funded by the Federal
Office of Meteorology and Climatology Meteo Swiss within the framework of GCOS Switzerland.

*Data Availability Statement.* The CERES SYN1deg data is available at https://ceres-tool.larc.nasa.gov/ord-
tool/jsp/SYN1degEd41Selection.jsp; The AIRS data is accessible from
https://disc.gsfc.nasa.gov/datasets/AIRS3STM_006/summary?keywords=AIRS; The MODIS data is from
https://ladsweb.modaps.eosdis.nasa.gov/archive/allData/61/MYD08_M3/?process=ftpAsHttp&path=allDat
a%2f61%2fMYD08_M3; The CloudSat data is from http://www.cloudsat.cira.colostate.edu/data-
products/level-2b/2b-cwc-ro; The MERRA-2 dataset is obtained at
https://disc.gsfc.nasa.gov/datasets/M2IMNPANA_5.12.4/summary?keywords=merra-2. The ERA-Interim
is from https://apps.ecmwf.int/datasets/data/interim-full-moda/levtype=sfc.

*Author contributions.* HZ, MW, and QW proposed the main ideas of this study. QW designed and wrote the
manuscript. SY provided the homogenized ground-based surface solar radiation data. QC, XZ, and GS
contributed to the interpretation of the results. BX and YW assisted with the figures. All co-authors
participated in discussions and provided constructive suggestions.

*Competing interests.* The authors declare that they have no conflict of interest.

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
