# Peer review of "An assessment of land energy balance over East Asia from multiple"

_Atmospheric Chemistry and Physics, 2022_

## Author Comment (AC1)

We first thank the very constructive comments of the reviewer. We have taken all of reviewer's comments into consideration and revised the manuscript accordingly. All the changes have been tracked in the revised manuscript. Our detailed responses are as follows.

**Response to Anonymous Referee #1**

This paper investigated the land energy balance over East Asia using several data, like surface observation data, satellite data, reanalysis data, and CMIP data. Results are interesting and indicate that a larger shortwave radiation of 5.2% is reflected and smaller shortwave absorption of 0.6% is estimated. In addition, cloud radiation effects (shortwave, longwave at the surface, atmosphere, and TOA) are also evaluated. Overall, this manuscript is clear. This study is of great significance to improve the new understanding of energy balance in East Asia. However, there are several issues that need to be taken care of before this paper becomes acceptable for publication.

Specific comments:

1. In Figure 6, the surface energy is not balanced due to the lack of sensible heat flux and latent heat flux values.

   Reply: The cloud-free energy budget shown in Figure 6b is not balanced because it is not the one that Earth would achieve in equilibrium when no clouds could form. It rather represents the mean fluxes as determined solely by removing the clouds but otherwise retaining the entire atmospheric structure. This enables the quantification of the effects of clouds on the Earth energy budget and corresponds to the way clear-sky fluxes are calculated in climate models. Thus, the cloud-free energy budget is not closed and therefore the sensible and latent heat fluxes are not quantified in Figure 6b. Please see the more comprehensive explanation in the revised manuscript.

2. In Figure 7, the Spatial distributions of annual mean SSR biases derived from CERES, CMIP6 and ERA5 are both overestimated in the high value region, please try to explain the reason.

Reply: I think this is not a common phenomenon only occurred over high value regions or in clear-sky conditions. It is very likely related to aerosols in clear-sky conditions, which is possibly attributed to the CMIP6 models' capabilities to represent the aerosol properties and retrieval algorithms of aerosols in satellite data over TP region.

3. The radiative effects and radiative forcing of aerosols are rarely discussed in this paper. How to distinguish the radiative effects of clouds and aerosols under the All-sky situation?

Reply: We used to distinguish the radiative effects of clouds and aerosols based on the surface solar radiation (SSR) under all-sky, clear-sky, all-sky-no-aerosol, and clear-sky-no-aerosol conditions derived from CERES SYN1deg product. Generally, clouds and aerosols dominate the all-sky SSR, whereas aerosols play a vital role in the clear-sky conditions. Please refer to Wang et al. (2021) for more detailed explanation on how to distinguish contributions from aerosols and clouds. Actually, it is not easy to distinguish the radiative effects of clouds and aerosols during the analysis of all-sky energy balance budget. We dealt with this here by combing the potential factors affecting the corresponding radiation components and their observational fact over East Asian and global land to determine the possible reasons for the discrepancies between them. This is shown in section 3.4.3 in the revised manuscript.

Reference:

Wang, Q., Zhang, H., Yang, S., Chen, Q., Zhou, X., Shi, G., Cheng, Y., and Wild, M.: Potential driving factors on surface solar radiation trends over China in

recent years, Remote Sens., 13, 704, https://doi.org/10.3390/rs13040704, 2021.

4. The authors emphasize that the fewer low clouds due to the TP are very likely the causes for the smaller fraction of East Asian land surface downward longwave radiation. Is the conclusion that there are fewer low clouds over the TP consistent with the actual situation?

Reply: Yes, it is. Please also see Figure S4 in the supplementary material, where the global distribution of land mean total and low cloud fraction (CF) during 2010-2014 derived from CERES_SSF1deg Ed4.1 product is also displayed. The corresponding area-weighted averages over East Asian land and global land are 56.3% and 55.2%, as well as 10.5% and 14.2%, respectively, suggesting a slightly more total CF of 1.1% and fewer low CF of 3.7% over East Asian land compared to the global land.

5. L573, why do you select ERA5 surface LW radiation as the reference? Why not choose CERES-EBAF as the reference?

Reply: Renalyses take into account possibly the best available estimates of atmospheric temperature and humidity profiles at high temporal frequency in their radiative transfer calculations, which further contributes to an accurate determination of the surface thermal radiation (Wild, 2017a). Wild et al. (2015) also reported that the best estimate of the downward thermal radiation is very close to the corresponding estimate of ERA-Interim reanalysis based on the various CMIP5 models and surface observational sites. However, the energy fluxes at the Earth surface cannot be directly measured from space, but have to be inferred from the measurable TOA signals using additionally empirical or physical models to remove atmospheric perturbations. Thus, they don't have the same accuracy as the TOA fluxes. Also, satellite-derived records of surface fluxes may also suffer from

potential inhomogeneities due to changes in satellites, viewing geometries, inaccurate positioning, or sensor degradation, particularly in the earlier records.

References:

Wild, M.: Progress and challenges in the estimation of the global energy balance, AIP Conference Proceedings, 1810, 20004, https://doi.org/10.1063/1.4975500, 2017a.

Wild, M., Folini, D., Hakuba, M. Z., Schär, C., Seneviratne, S. I., Kato, S., Rutan, D., Ammann, C., Wood, E. F., and König-Langlo, G.: The energy balance over land and oceans: an assessment based on direct observations and CMIP5 climate models, Clim. Dynam., 44, 3393–3429, https://doi.org/10.1007/s00382-014-2430-z, 2015.

6. It is better to introduce relative research (e.g., Li et al., Xu et al., Letu et al. 2022) in the introduction part.

Li, M., Letu, H., Peng, Y., Ishimoto, H., Lin, Y., Nakajima, T. Y., ... & Shi, J. (2022). Investigation of ice cloud modeling capabilities for the irregularly shaped Voronoi ice scattering models in climate simulations. Atmospheric Chemistry and Physics, 22(7), 4809-4825.

Xu, J., Liang, S., & Jiang, B. (2022). A global long-term (1981–2019) daily land surface radiation budget product from AVHRR satellite data using a residual convolutional neural network. Earth System Science Data, 14(5), 2315-2341.

Letu, H., Nakajima, T. Y., Wang, T., Shang, H., Ma, R., Yang, K., ... & Shi, J. (2022). A new benchmark for surface radiation products over the East Asia–Pacific region retrieved from the Himawari-8/AHI next-generation geostationary satellite. Bulletin of the American Meteorological Society, 103(3), E873-E888.

Reply: Thanks for the good suggestion. Some of the related publications are cited in the end of the manuscript for possible comparisons in the future. The sentence is added as "For example, newly published surface radiation products with high resolutions based on satellite datasets (e.g., Letu et al., 2022; Xu et al., 2022) are expected to make sense in improving the accuracy of the regional/global surface radiation budget studies.".

---

## Author Comment (AC2)

We first thank the very constructive comments of the reviewer. We have taken all of reviewer's comments into consideration and revised the manuscript accordingly. All the changes have been tracked in the revised manuscript. Our detailed responses are as follows.

**Response to Anonymous Referee #2**

This paper assessed the land energy balance over East Asia with surface measurements, satellite estimations, reanalysis and CMIP6 products. The author found a substantially larger fraction of atmospheric shortwave radiation of 5.2% is reflected and a slightly smaller fraction of atmospheric shortwave absorption of 0.6%. The author also investigated the cloud radiative effects and found the presence of clouds results in a larger cooling effect on the climate system over East Asian land than that over globe. The author also pointed out the role of Tibet Plateau in both SW and LW radiation balance.

The paper addresses an important and interesting topic, and I believe that this paper have the potential to be of great value to the scientific community. However, I have several concerns as outlined below. After addressing these concerns, I believe the work would be a good fit for publication.

1. The topic of this paper is land energy balance over East Asia and the role of TP, aerosols and clouds. The author analyzed the cloud radiative effects in section5. However the author did not outlined the importance to study cloud radiative effects. I suggest to expand the discussion on the cloud radiative effects in introduction to emphasize the importance of authors analysis.
   Reply: Really appreciate for the constructive suggestion. The discussion of cloud radiative effects in the introduction part is added as "Moreover, clouds play a key role in modulating global and regional energy budgets and hydrological cycles through increasing the reflected solar radiation and also the downward thermal

radiation, leading to a cooling and warming of climate system (Stephens, 2005; Wild et al., 2013a; Li et al., 2015; H. Wang et al., 2021).".

References:

Li, J., and Mao, J.: A preliminary evaluation of global and East Asian cloud radiative effects in reanalyses, Atmos. and Ocean. Sci. Lett., 8, 100-106, https://doi.org/10.3878/AOSL20140093, 2015.

Stephens, G. L.: Cloud feedbacks in the climate system: A critical review, J. Climate, 18, 237-273, https://doi.org/10.1175/JCLI-3243.1, 2005.

Wang, H., Zhang, H., Xie, B., Jing, X., He, J., and Liu, Y.: Evaluating the Impacts of Cloud Microphysical and Overlap Parameters on Simulated Clouds in Global Climate Models, Adv. Atmos. Sci., https://doi.org/10.1007/s00376-021-0369-7, 2021.

Wild, M., Folini, D., Schär, C., Loeb, N., Dutton, E. G., and König-Langlo, G.: The global energy balance from a surface perspective, Clim. Dynam., 40, 3107–3134, https://doi.org/10.1007/s00382-012-1569-8, 2013a.

2. This paper is lack of the discussion on the radiative effects of aerosols.

Reply: Yes, it is. Actually, it is not easy here to expand detailed discussion on the aerosol radiative effects. We only cited the most related reference documented by Wei et al. (2019), who provided the statistics of regional and global land mean AODs from MODIS aerosol products. On this basis, we tried to using the fact of more aerosol loadings over East Asian land to explain the potential causes contributing to the discrepancies in the estimated global and East Asian land energy budgets as shown in section 3.4.3.

Reference:

Wei, J., Peng, Y., Guo, J., and Sun, L.: Performance of MODIS Collection 6.1
Level 3 aerosol products in spatial-temporal variations over land, Atmos.
Environ., 206, 30–44, https://doi.org/10.1016/j.atmosenv.2019.03.001, 2019.

3. Why the surface sites are divided into rural/urban categories? Is this related to
surface properties or aerosol or both? Please explain.

Reply: Yes, two factors of surface properties and aerosol emissions are certainly
considered in choosing the rural and urban sites. Surface properties mainly
influence reflected solar radiation at the surface by affecting their albedos, while
aerosols can exert a strong influence on shortwave radiation. Considering the
urbanization effect on surface solar radiation in China is still disputed (e.g., Wang
et al., 2014; Imamovic et al., 2016), an attempt to quantify the anthropogenic
influence on surface solar radiation through dividing surface sites into urban/rural
categories is particularly necessary and meaningful in this case. Hopefully this in-
depth exploration would also provide an opportunity to examine the performances
of different products (such as satellite retrievals, climate models, and reanalysis) to
derive surface solar radiation at urban/rural sites.

References:

Wang, K., Ma, Q., Wang, X., and Wild, M.: Urban impacts on mean and trend of
surface incident solar radiation, Geophys. Res. Lett., 41, 4664–4668, 2014.

Imamovic, A., Tanaka, K., Folini, D., and Wild, M.: Global dimming and
urbanization: did stronger negative SSR trends collocate with regions of
population growth? Atmos. Chem. Phys., 16, 2719–2725,
https://doi.org/10.5194/acp-16-2719-2016, 2016.

4. Table 1, the land-atmosphere net heat flux is worth known.

Reply: Referring to the published energy balance assessment papers (e.g., Wild et

al., 2015; Wild, 2020), all the estimated energy balance components as well as the CREs from the TOA, within the atmosphere, and at the surface under both all-sky and clear-sky conditions have been given in Table 1. Thanks for the suggestion.

References:

Wild, M.: The global energy balance as represented in CMIP6 climate models, Clim. Dynam., 55, 553–577, https://doi.org/10.1007/s00382-020-05282-7, 2020.

Wild, M., Folini, D., Hakuba, M. Z., Schär, C., Seneviratne, S. I., Kato, S., Rutan, D., Ammann, C., Wood, E. F., and König-Langlo, G.: The energy balance over land and oceans: an assessment based on direct observations and CMIP5 climate models, Clim. Dynam., 44, 3393–3429, https://doi.org/10.1007/s00382-014-2430-z, 2015.

5. Table 2, 17.2 W m$^{-2}$ could be a large difference, the percentage-wise is worth known.

   Reply: Agreed. The largest percentages of SSR biases relative to their respective station-mean averages are estimated to be around 10% and 4% for all-sky and clear-sky conditions. Please see the updated caption of Table 2 in the revised manuscript.

6. Line602-613, "Comparisons…..OLR of 2.7%", I suggest to show the spatial distribution of CRE from reanalysis or satellite estimations or CMIP6 over East Asia in section 5 and the spatial distribution of net heat flux (from ERA5 or CMIP6) to support this conclusion on the role of TP.

   Reply: As the world's largest and highest plateau, TP accounts for nearly one third of the East Asian land area, which shows lower water vapor content effects for shortwave radiation and weaker greenhouse effects for downward thermal radiation due to its orographic effect over East Asian land compared to the global

land estimates. Thus, I think it is not necessary to also give the detailed spatial distributions of CREs and net heat flux (I guess the 'net heat flux' here represents the surface net flux?) for the further explanation. Besides, the limited length of the article is also considered here. Thanks for the suggestion.

---

## Author Response (AR2)

We first thank the very constructive comments of the reviewers. We have taken all the reviewers' comments into consideration and revised the manuscript accordingly. All the changes have been tracked in the revised manuscript. Our detailed responses are as follows.

**Response to Anonymous Referee #1**

This manuscript describes the land energy balance over East Asia with surface measurements, satellite estimations, reanalysis and CMIP6 products. The paper finds that the high aerosol loadings, clouds, and the Tibet Plateau over East Asia play vital roles in the regional shortwave budgets. the And Tibet Plateau is also responsible for the longwave budgets. The cloud radiative effects over East Asia are also assessed by this paper. The author pointed out that the presence of clouds results in a larger cooling effect on the land climate system over East Asian than that over globe. I believe that it is important to understand the land energy over East Asia. And this paper has the potential to be of great value to the scientific community.

Comments:

1. Line 155: "the first ensemble member" refers to the control member?

   Reply: No, "the first ensemble member" refers to the first ensemble member named r1i1p1f1 for each model. The CMIP6 "historical all forcings" experiments used in this study include several ensemble members for each radiative component, such as r1i1p1f1, r1i1p1f2 etc., and r1i1p1f1 is comprised of the largest available models.

2. Line 160: The link needs to direct to the right product

   Reply: Accepted. The link to ERA5 dataset is updated to a new link (https://cds.climate.copernicus.eu/cdsapp#!/dataset/reanalysis-era5-single-levels-monthly-means?tab=form). Thanks for your correction.

3. Line 162-163: It is better to use "as well as a higher vertical resolution with ?? levels"

   Reply: Accepted. This sentence is revised to "as well as a higher vertical resolution with 137 levels".

4. Line 167-171: ERA5 has different performance at different locations. what about the bias of ERA5 over East Asia. Please point out the purpose of using ERA5 in this paper?

   Reply: ERA5 provides estimates of surface radiation fluxes with higher resolution compared to its predecessor ERA-Interim, and the accuracies of which should also be greatly improved because the new product assimilates more satellite signals and more station data. Compared to the satellite-derived product SARAH, ERA5 shows the smallest deviations for most Europe, Central Asia and South Africa in terms of surface irradiance, while the largest positive deviations are observed in the Tibetan Plateau and China. This overestimation in China is possibly related to an underestimation of clouds and anthropogenic aerosols as documented by Zhang et al. (2016). To sum up, the estimated surface irradiances from ERA5 are comparable with the corresponding satellite-based estimates over most inland regions with low occurrence of clouds (Urraca et al., 2018). However, for surface downward longwave radiation, Tang et al. (2021) pointed out that the accuracy of the ERA5 over land is higher than CERES-derived product on average against BSRN station data both at hourly and monthly time scales. Moreover, since the reanalyses assimilate the observed state of the atmosphere several times per day, they are considered as a useful source for the determination surface upward longwave radiation (Wild, 2017a). Thus, the purpose of using ERA5 in this paper is for the best estimations of the surface longwave radiation.

References:

Tang, W., Qin, J., Yang, K., Zhu, F., and Zhou, X.: Does ERA5 outperform satellite products in estimating atmospheric downward longwave radiation at the surface? Atmos. Res., 252, 105453, https://doi.org/10.1016/j.atmosres.2021.105453, 2021.

Urraca, R., Huld, T., Gracia-Amillo, A., Martinez-de-Pison, F. J., Kaspar, F., and Sanz-Garcia, A.: Evaluation of global horizontal irradiance estimates from ERA5 and COSMO-REA6 reanalyses using ground and satellite-based data, Sol. Energy, 164, 339–354, https://doi.org/10.1016/j.solener.2018.02.059, 2018.

Wild, M.: Progress and challenges in the estimation of the global energy balance, AIP Conference Proceedings, 1810, 20004, https://doi.org/10.1063/1.4975500, 2017a.

Zhang, X., Liang, S., Wang, G., Yao, Y., Jiang, B., Cheng, J.: Evaluation of reanalysis surface incident shortwave radiation products from NCEP, ECMWF, GSFC, and JMA, using satellite and surface observations. Remote Sens., 8 (3), 225, http://dx.doi.org/10.3390/rs8030225, 2016.

5. L195-200: Why the CIESM is kept in Figure 1, It does not have value for all conditions.

   Reply: The reason for keeping the CIESM model in the figures is that this model has values for latent and sensible heat as presented in Figure S2 in supplemental material.

6. L195: Any comments on why some models have very large anomalies. For example, the canESM5 has very large anomalies for surface net SW for both All-sky & Clear sky, any possible reasons?

   Reply: As shown in the following figures, the large anomalies of surface net SW

radiation both under all-sky and clear-sky conditions for some models are highly related to their corresponding surface downward and upward SW radiation. For example, the large anomaly of all-sky surface upward radiation in canESM5 model compared to multi-model mean (-12 W m$^{-2}$) is the major contributor to the large anomaly in surface net SW radiation (Figure R1), which is possibly associated with its higher surface albedo compared to other CMIP6 models. For clear-sky conditions, both the much lower and higher surface downward and upward SW radiation (-14 and -11 W m$^{-2}$) compared to their multi-model means contributed to the large anomaly (-25 W m$^{-2}$) in surface net SW radiation (Figure R2). This is likely linked to the higher simulated aerosol loadings and surface albedo in canESM5 model with respect to the multi-model mean levels.

[Figure]

[Figure]

**Figure R1**. Annual land mean surface downward, upward, and net shortwave radiation (Units: W m$^{-2}$) under all-sky conditions over East Asia as simulated by various CMIP6 models (orange bars). The corresponding estimates from CMIP6 multi-model mean, CERES-EBAF and ERA5 are also presented in black, blue, and red bars.

[Figure]

[Figure]

**Figure R2**. Same as figure R1, but for clear-sky conditions.

7. Line 252-254: The sample sizes of the rural/urban are not same. And if the sample size is very small, the statistics could be highly affected by extreme large/small values. It is better to give the percentage of rural/urban sites that have positive/negative bias.

Reply: Thanks for the suggestion. We have updated Table 2 with their corresponding SSR bias percentages.

| Station-mean SSR biases | All-sky | | | Clear-sky | | |
|---|---|---|---|---|---|---|
| (Unit: W m$^{-2}$) | all | urban | rural | all | urban | rural |
| CERES-EBAF | 3.8 (2.3%) | 4.2 (2.6%) | 1.7 (0.9%) | 0.4 (0.2%) | 0.5 (0.2%) | -0.3 (-0.1%) |
| CMIP6 | 13.8 (8.3%) | 15 (9.2%) | 7.4 (4.1%) | 9.1 (4%) | 9.7 (4.3%) | 6.4 (2.8%) |
| ERA5 | 16.5 (10%) | 17.2 (10.5%) | 12.7 (7%) | 5.7 (2.5%) | 6.2 (2.7%) | 3.6 (1.5%) |

8. Line 514-516: "All the uncertainty ranges …. for their TOA counterparts" This sentence is confusing, please reformat it.

Reply: Accepted. This sentence is revised as "Apart from the TOA budget, all the rest uncertainty ranges are given by different data sources from various CMIP6 models, as well as the multi-model mean, CERES-, and ERA5-derived estimates."

9. Line 572-573 "The above CRE estimates are compared to the corresponding estimates from different data sources" This sentence is confusing. I think that is "The above CRE best estimates are compared with the corresponding estimates from different data sources". Please pay attention to use the phrase of "compared to" & "compared with"

Reply: Accepted. This sentence has been revised. Thanks for the suggestion.

10. Line 588-593: The same as figure 1, the CIESM does not have values for all conditions. Why it is kept in the figure 9?

Reply: The reason for keeping the CIESM model in the figures is that this model has values for latent and sensible heat as presented in Figure S2 in supplemental material.

**Response to Anonymous Referee #2**

The authors have addressed my comments, and the revised version reads much better.

One minor question:

1. Based on the results of Fig9, it is recommended to answer the following two questions:why GISS-E2-1-G, KACE-1-0-G show the maximum negative anomalies? Why INM-CM4-8 has the maximum positive anomalies?

Reply: As shown in the following figures, the large negative TOA SW CRE in GISS-E2-1-G model contributes to the maximum negative TOA net CRE (Figure R1), while both large negative SW and LW CREs within the atmosphere result in the maximum negative atmospheric CREs in the model (Figure R2). The large negative and positive SW CREs within the atmosphere are the major contributors to the maximum negative and positive atmospheric net CREs in KACE-1-0-G and INM-CM4-8 models (Figure R2), respectively. The smallest negative SW CRE at the surface in KACE-1-0-G model is the major cause for the minimum negative surface net CRE among the models (Figure R3). The reasons for the large discrepancies in the above models are highly related to these models' representations of aerosol and clouds as well as the surface albedo.

[Figure]

[Figure]

**Figure R1**. Annual land mean SW, LW, and net (SW+LW) CREs (Units: W m$^{-2}$) at the TOA over East Asia as simulated by various CMIP6 models (orange bars). The corresponding estimates from CMIP6 multi-model mean, CERES-EBAF and ERA5 are also presented in black, blue, and red bars.

[Figure]

[Figure]

[Figure]

**Figure R2**. Same as figure R1, but for CREs within the atmosphere.

[Figure]

[Figure]

**Figure R3**. Same as figure R1, but for CREs at the surface.